# Current Management of Locally Advanced Esophageal and Esophagogastric Junction Cancers: Clinical Evidence and Evolving Strategies

**DOI:** 10.3390/cancers17223603

**Published:** 2025-11-08

**Authors:** Andrea Di Donato, Marc Van den Eynde

**Affiliations:** Department of Medical Oncology and Gastroenterology, Cliniques Universitaires St-Luc - Institut Roi Albert II, Université Catholique de Louvain, 10 Av. Hippocrate, 1200 Bruxelles, Belgium; andrea.didonato@student.uclouvain.be

**Keywords:** esophageal cancer, esophagogastric junction cancer, perioperative treatment, immunotherapy, targeted therapy, radiotherapy

## Abstract

Despite improvements in multimodal therapies, locally advanced esophageal and esophagogastric junction cancers still carry a high risk of recurrence and remain difficult to control long term. This review summarizes current treatment strategies, which are tailored according to tumor type. In adenocarcinomas, perioperative chemotherapy combined with immunotherapy has become the preferred approach following results from recent clinical trials. Meanwhile, chemoradiotherapy remains the mainstay for esophageal squamous cell carcinoma. New strategies are being explored to personalize treatment, including the use of biomarkers like MSI for select patients who might benefit from immunotherapy and organ-preserving approaches for complete responders to reduce unnecessary surgery. This review provides an overview of these advances and the ongoing efforts to improve outcomes in localized disease.

## 1. Introduction

Cancers of the esophagus and the esophagogastric junction (EGJ) represent a significant global health challenge, ranking among the most lethal malignancies worldwide. With an estimated over 600,000 new cases annually, esophageal cancer remains a highly lethal disease [1]. While overall 5-year survival rates reflecting all disease stages rarely exceed 20–25% for esophageal squamous cell carcinoma (SCC) and are slightly higher for adenocarcinoma (AC) in Western countries [2], outcomes are significantly better in the subset of patients with resectable locally advanced tumors treated with multimodal approaches, with 10-year survival rates reaching 38% in recent landmark trials [3].

The histological dichotomy between SCC and AC reflects not only distinct etiopathogeneses of tobacco and alcohol for SCC, and obesity and gastroesophageal reflux for AC, but also diverging anatomical location (proximal for SCC and distal for AC) and therapeutic sensitivities [4]. While SCC still accounts for the majority of esophageal cancers worldwide, its proportion has declined in recent years, with recent Chinese registry data showing that SCC now represents less than 90% of cases [5,6]. Meanwhile, AC has become more common than SCC in parts of Europe, North America, and some high-risk Asian regions, a shift associated with economic growth and lifestyle changes.

EGJ adenocarcinoma further complicates management decisions due to their anatomical ambiguity and overlapping treatment paradigms between gastric and esophageal cancers. The molecular heterogeneity of these tumors is increasingly recognized, with The Cancer Genome Atlas (TCGA) identifying subtypes based on chromosomal instability, deficient mismatch repair (dMMR)/Microsatellite Instability-High (MSI-H), Epstein–Barr virus (EBV) status, and genomic stability, paving the way for a more personalized therapeutic approach [7].

Over the past decade, the standard of care for locally advanced esophageal and EGJ cancers (cT2–4a, cN+/–, cM0) has shifted from surgery alone to multimodal strategies, including neoadjuvant chemoradiotherapy [3,8,9], perioperative chemotherapy [10,11,12,13,14], and more recently, adjuvant or perioperative immunotherapy [15,16,17]. However, multiple questions remain unresolved, particularly regarding the optimal sequence and combination of therapies across histological subtypes, the role of current and emerging biomarkers used in metastatic settings, and the feasibility of non-operative strategies in selected patients. Recent trials are redefining the landscape by integrating immune checkpoint inhibitors [15,16,18], refining chemotherapy backbones [10,19], and assessing biomarker-driven selection [20,21]. Furthermore, the advent of liquid biopsies such as circulating tumor DNA (ctDNA) and advanced imaging could enable real-time response adaptation [22].

This review aims to provide an up-to-date synthesis of current and emerging treatment strategies for locally advanced esophageal and EGJ cancers. Emphasis will be placed on the impact of histology, molecular profiling, and the evolving role of immunotherapy and targeted therapies.

## 2. The Importance of an Adequate Tumor Staging

Accurate diagnostic and staging work-up is a critical step in the management of esophageal and EGJ cancers. Given the diverse biological behaviors of SCC and AC, as well as the anatomical complexity of the EGJ, a multimodal staging approach is required to optimally guide treatment decisions [23].

Initial diagnosis relies on high-definition endoscopy combined with systematic biopsies to confirm histology and to characterize tumor location especially for EGJ lesions categorized under the Siewert classification [24]. For early-stage tumors (cT1), enhanced imaging modalities such as narrow-band imaging (NBI) and chromoendoscopy improve the detection of superficial lesions and guide decisions on endoscopic resectability [23,25].

Staging must include a complete clinical examination, a contrast-enhanced computed tomography (CT) of the chest, abdomen and pelvis and/or a [18F]2-fluoro-2-deoxy-D-glucose (FDG) positron emission tomography (PET). FDG-PET, particularly helpful in identifying undetected distant metastatases, should be carried out in patients who are candidates for esophagectomy [26]. Endoscopic ultrasound (EUS) is useful for tumor (T) and nodal (N) staging, but its accuracy is limited for T1 tumors where endoscopic resection provides more reliable staging while also offering therapeutic benefit [23,25]. EUS helps guide therapy by assessing T4b invasion of adjacent structures (airways, pericardium or aorta) and by identifying or sampling lymph node metastases beyond standard treatment fields. In advanced T stages, strictures may limit its use, in which case bronchoscopy with endobronchial ultrasound is a valuable alternative, particularly for evaluating airway involvement [23,25]. For locally advanced (cT3/T4) EGJ AC infiltrating the anatomical cardia (Siewert II-III), staging laparoscopy with peritoneal cytology is recommended to detect occult peritoneal metastases found in 15% of patients [23,27].

Head and neck second primary tumors occur in about 6.7% of patients with esophageal SCC and are associated with poorer prognosis [28]. Early detection through thorough head and neck examination is therefore essential to improve outcomes.

Esophageal and EGJ cancers should be staged according to the American Joint Committee on Cancer (AJCC)/Union for International Cancer Control (UICC) tumor- node-metastasis (TNM) 8th edition staging system to refine prognostic stratification [22,24]. This staging framework aligns with treatment algorithms that are increasingly histology-specific.

Unlike metastastatic disease, the molecular classification of locally advanced tumors currently remains less useful. The most impactful marker identified so far is dMMR/MSI-H [29]. Though rare (3–7%), when present, it opens the door to immunotherapy strategies that could fundamentally alter treatment sequences and modality even raising the possibility of non-surgical approaches in selected patients with complete response [20,30]. The role of HER-2 (for AC) and programmed cell death-ligand1 (PD-L1) biomarkers (both histologies) are currently less relevant.

## 3. Current Management of Locally Advanced Disease

### 3.1. SCC Esophageal Cancer

#### 3.1.1. Multimodal Approach Including Surgery

For locally advanced esophageal SCC, surgery alone has been consistently associated with poor long-term outcomes, with 5-year survival rates rarely exceeding 30% [31]. Table 1 summarizes key randomized trials that have shaped the multimodal management of locally advanced esophageal SCC, including neoadjuvant chemoradiotherapy (nCRT), perioperative strategies, and active surveillance. The landmark CROSS trial, which enrolled both SCC (23%) and AC (75%) tumors, demonstrated a significant improvement in R0 resection rate (92% vs. 69%) and median overall survival (overall survival (OS): 49.4 vs. 24.0 months; HR = 0.66, *p* = 0.003) in the nCRT arm (weekly carboplatin and paclitaxel, 41.4 Gy, 5 weeks of treatment), with even greater benefit observed in SCC patients (10-year OS rate: 46% vs. 23%; *p* = 0.007) than in AC patients (10-year OS: 36% vs. 23%; *p* = 0.061) [3]. In contrast, the Fédération Francophone de Cancérologie Digestive (FFCD) 9901 trial enrolling patients with less advanced tumors (stage I–II, 70.3% SCC) showed that nCRT did not offer any survival benefit compared to surgery alone, but it increased postoperative mortality (11.1% vs. 3.4%; *p* = 0.049) [31].

**Table 1 cancers-17-03603-t001:** Key trials evaluating multimodal approach in locally advanced esophageal SCC.

Study	Population	Intervention Arm	Control Arm	Primary Endpoint	Outcomes
CROSS (phase III) [3,8]	*n* = 366 SCC (23%)-Gastric/EGJ AC (75%) Stage II–IIIa	nCRT (Carboplatin/paclitaxel) + RT 41.4 Gy in 23 daily fractions + surgery	Surgery alone	OS	- mOS: 49.4 vs. 24.0 months (HR 0.657, *p* = 0.003) - 10-year OS (SCC subgroup): 46% vs. 23% (*p* = 0.007) - R0 resection 92% vs. 69% (*p* < 0.001) - pCR (nCRT group): 49% for SCC vs. 23% for AC
FFCD 9901 (phase III) [31]	*n* = 195 SCC (70.3%)–Gastric/EGJ AC (29.2%) Stage I–II	nCRT (Fluorouracil/Cisplatin) + RT 45 Gy: 25 fractions over 5 weeks + surgery	Surgery alone	OS	- 3-year OS: 47.5% vs. 53% (HR 0.99; *p* = 0.94) - R0 resection: 93.8% vs. 92.1% (*p* = 0.749) - Postoperative mortality: 11.1% vs. 3.4% (*p* = 0.049) - 5-year DFS: 35.6% vs. 27.7% (*p* = 0.65)
JCOG9907 (phase III) [32,33]	*n* = 330 SCC (100%) Stage II or III (excluding T4)	Surgery + Postop CT (Cisplatin/ 5-FU)	Preop CT(Cisplatin/ 5-FU) + Surgery	PFS	- 3-year PFS: 44.2% vs. 55.8% (HR 1.38; *p* = 0.16) - 5-year OS: 43% vs. 55% (HR = 0.73; *p* = 0.04)
JCOG1109 NExT (phase III) [34]	N = 601 SCC (99%) Stage Ib–III (excluding T4)	NeoDCF + surgery NeoCF + RT + surgery	NeoCF + surgery	OS	- 3-year OS NeoDCF vs. NeoCF: 72.1% vs. 62.6% (HR = 0.68; *p* = 0.006) - 3-year OS NeoCF + RT: 68.3% (HR = 0.84; *p* = 0.12) - 3-year PFS neoDCF vs. neoCF: 61.8% vs. 47.7% (HR 0.67; *p*: NA) - 3-year PFS NeoCF + RT: 58.5% (HR 0.77; *p*: NA)
CheckMate 577 (phase III) [16,35]	*n* = 794 SCC (29%) - Gastric/EGJ AC (71%) Stage II–III	nCRT + surgery + adjuvant Nivolumab	nCRT + surgery + adjuvant placebo	DFS	- mDFS: 21.8 months vs. 10.8 months (HR 0.76; *p* < 0.001) SCC subgroup: HR for DFS: 0.61 - mOS: 51.7 m vs. 35.3 m (HR: 0.85; *p* = 0.1064) SCC subgroup: HR: 0.72 If CPS ≥ 1: mOS: 45.5 months vs. 33.5 months (HR: 0.79) If CPS < 1: mOS: 39.2 months vs. 52.8 months (HR: 1.40)
RTOG 85-01 (phase III) [36]	*n* = 192 (randomized part: *n* = 123) (nonrandomized part: *n* = 69)) SCC – Gastric/EGJ AC Stage I–IIIa	RT 50Gy in 25 fractions over 5 weeks + CT (Cisplatin/fluorouracil)	RT alone 64Gy in 32 fractions over 6.4 weeks	OS	- 5-year OS: 26% (randomized CMG) vs. 14% (nonrandomized CMG) vs. none (RT group) (*p* = 0.24)
ARTDECO (phase III) [37]	*n* = 260 SCC (61%) - Gastric/EGJ AC (39%) Stage II–IVa	HD of 61.6 Gy + CT (Carboplatin/paclitaxel)	SD of 50.4 Gy/1.8 Gy for 5.5 weeks + CT (Carboplatin/paclitaxel)	LPFS	- 3-year LPFS: 73% vs. 70% (*p* = 0.62) - 3-year locoregional PFS: 59% vs. 52% (*p* = 0.08) - 3-year OS: 77% vs. 60% (*p* = 0.12) - Grade 4 and 5 toxicity: 14% and 10% (for HD) vs. 12% and 5% (for SD) (*p* = 0.15)
PRODIGE-5/ACCORD17 (phase II-III) [38]	*n* = 267 SCC (85%) – Gastric/EGJ AC (14%) Stage I–IVa	FOLFOX + RT 50 Gy in 25 fractions	5-FU/Cisplatin 75 mg/m^2^ + RT 50 Gy in 25 fractions	PFS	- mPFS: 9.7 months vs. 9.4 months (HR: 0.93; *p* = 0.64) - mOS: 20.2 months vs. 17.5 months (HR = 0.94; *p* = 0.70) - Toxic death: 1 case vs. 6 (*p* = 0.066)
SANO trial (phase III) [9,39]	*n* = 309 SCC (24%) – Gastric/EGJ AC (74%) Stage Ib–III	nCRT (carboplatin/paclitaxel + RT 41.4 Gy in 23 fractions + Active surveillance	nCRT (carboplatin/paclitaxel + RT 41.4 Gy in 23 fractions + Surgery	OS	- 2-year OS: 74% vs. 71% (HR 0.83; *p* = 0.42) - mDFS: 35 months vs. 49 months (HR 1.25; *p* = 0.29)

Abbreviations: Adenocarcinoma (AC), Chemotherapy (CT), Chemoradiotherapy (CRT), Combined Modality Group (CMG), Disease-Free Survival (DFS), Esophagogastric Junction (EGJ), Hazard Ratio (HR), High Dose (HD), local Progression-Free Survival (LPFS), median Disease-Free Survival (mDFS), median Overall Survival (mOS), median Progression-Free Survival (mPFS), neoadjuvant doublet chemotherapy consisting of fluorouracil and cisplatin (neoCF), definitive chemoneoadjuvant triplet chemotherapy consisting of fluorouracil, cisplatin, and docetaxel (neoDCF), neoadjuvant doublet chemotherapy consisting of fluorouracil and cisplatin with radiotherapy (NeoCF + RT), Not Available (NA), Overall Survival (OS), pathologic Complete Response (pCR), Programmed Death Ligand 1 (PD-L1) Progression-Free Survival (PFS), Preoperative (Preop), Postoperative (Postop), Radiotherapy (RT), Squamous Cell Carcinoma (SCC), Standard Dose (SD).

The Asian Japan Clinical Oncology Group (JCOG)9907 trial compared neoadjuvant chemotherapy (cisplatin + 5-FU) to postoperative chemotherapy in patients with stage II/III SCC and demonstrated a significant improvement in OS in favor of the neoadjuvant approach (5-year OS: 55% vs. 43%; HR = 0.73; *p* = 0.04). Although the primary endpoint of 3-year (progression-free survival) PFS did not reach statistical significance (55.8% vs. 44.2%; HR = 1.38; *p* = 0.16), the survival benefit supported the shift toward preoperative chemotherapy in this population [32,33]. The role of radiotherapy in neoadjuvant settings continues to be debated, particularly in Asia. The Japanese JCOG1109 (NExT) trial investigating treatment intensification with triplet chemotherapy (DCF: docetaxel, ciplastin, fluorouracil) or nCRT (41.4 Gy, combining CF) compared to neoadjuvant doublet chemotherapy (CF) reported that neoadjuvant triplet chemotherapy followed by esophagectomy resulted in a significant OS benefit compared with doublet chemotherapy (3-year OS: 72.1% vs. 62.6%; HR = 0,68; *p* = 0.006) [34]. nCRT did not show significant improvement of OS compared with doublet chemotherapy (CF). While these results suggest neoadjuvant chemotherapy with DCF is the standard of care for locally esophageal SCC in Asia, nCRT remains the preferred approach in many Western and international guidelines.

Efforts have shifted toward optimizing neoadjuvant strategies, particularly the integration of immunotherapy. Recent phase III trial have explored combining immunotherapy with nCRT or chemotherapy in SCC. The phase III CheckMate 577 trial demonstrated a significant benefit in disease-free survival (DFS) with adjuvant nivolumab (anti-PD-1) in patients with residual disease after nCRT and surgery (22.4 vs. 11.0 months; HR = 0.69; *p* < 0.001), with a stronger effect in the SCC subgroup (HR = 0.61). OS, though not statistically significant in the overall population, favored nivolumab numerically (median OS (mOS): 51.7 vs. 35.3 months; HR = 0.85; *p* = 0.1064), with a more pronounced effect observed in key subgroups [34]. Notably, patients with PD-L1 combined positive score (CPS) ≥ 1 experienced a significant OS improvement (45.5 vs. 33.5 months; HR = 0.79), while those with CPS < 1 did not derive benefit (39.2 vs. 52.8 months; HR = 1.40). Similarly, benefit was greater in SCC (HR = 0.72) [16,35]. These findings consolidate adjuvant nivolumab for patients with residual disease following trimodal therapy, particularly in PD-L1–positive or SCC populations.

#### 3.1.2. Multimodal Approach Without Surgery

In patients with esophageal SCC who are not resectable (cT4b with invasion of adjacent structures) or are unfit for surgery due to comorbidities or advanced age, or those who actively seek organ preservation, definitive chemoradiotherapy (dCRT) has long been considered an established alternative [36]. Based on early pivotal studies such as Radiation Therapy Oncology Group (RTOG) 85-01, dCRT combining cisplatin and fluorouracil with 50.4 Gy radiotherapy provided a survival benefit over radiotherapy alone, establishing it as a standard non-surgical modality. However, long-term outcomes remained suboptimal, with 5-year OS below 30% and frequent loco-regional recurrences.

The addition of induction chemotherapy or dose escalation in radiotherapy has not shown consistent benefit [40]. For instance, the PRODIGE 26/CONCORDE trial, which tested dose-escalated RT (66 Gy vs. 50.4 Gy) with concurrent chemotherapy (FOLFOX-4 for 3 courses followed by 3 adjuvant courses) in SCC, showed no significant OS improvement (HR 0.99; *p* = 0.94), while increasing toxicity [41]. Similarly, the ARTDECO trial evaluating radiation dose escalation up to 61.6 Gy to the primary tumor combined with weekly carboplatin-paclitaxel administration did not report a significant increase in local control over 50.4 Gy [37]. The absence of a dose effect was observed in both AC and SCC. Notably, the high-dose arm was associated with increased grade 4–5 toxicity (14% and 10% for high-dose arm vs. 12% and 5% for standard-dose arm), although this difference was not statistically significant (*p* = 0.15). Thus, the standard dose remains 50.4 Gy. The PRODIGE-5/ACCORD-17 trial reported that FOLFOX chemotherapy regimen might be a more convenient option to combine within dCRT compared to cisplatin–5FU, with similar efficacy both in terms of PFS (mPFS 9.7 vs. 9.4 months; HR 0.93; *p* = 0.64) and OS (mOS 20.2 vs. 17.5 months; HR 0.94; *p* = 0.70) [38].

Recent years have seen a paradigm shift toward integrating immunotherapy into dCRT, leveraging the immune-stimulatory effects of radiation. The ongoing KEYNOTE-975 phase III trial currently evaluates the addition of pembrolizumab (anti-PD-1) to dCRT in locally advanced esophageal cancer (SCC and AC), with OS and PFS as co-primary endpoints [42]. Earlier phase trials have shown encouraging results [43,44]. For example, the phase 1 PALACE-1 evaluated concurrent durvalumab with dCRT in unresectable SCC, reported a 1-year OS rate of 85.6% and a 2-year OS of 74.3%, with manageable toxicity [43]. However, the lack of randomization and small sample size precludes definitive conclusions.

Another key aspect is the organ-preservation strategy being investigated in operable patients. The phase III SANO trial, conducted in the Netherlands and including mixed histology (24% SCC), evaluated active surveillance after nCRT in patients with clinical complete response (cCR) based on endoscopic biopsies, EUS, and FDG-PET/CT [8,39]. This study shows that after a minimum follow-up of 2 years, OS was non-inferior after active surveillance compared with standard surgery (3-year OS: 84% vs. 82%, HR 0.92, *p* = 0.67), and it resulted in better health-related quality of life suggesting that deferring esophagectomy may be feasible in highly selected patients. A similar approach is being explored in the French PREACT trial and the Dutch NEEDS trial [45,46]. Importantly, the proportion of patients achieving cCR and the risk of missed residual disease are major limitations, underscoring the need for robust response assessment tools.

As trials mature, future data will clarify whether immunotherapy-enhanced dCRT or active surveillance post-nCRT can safely supplant surgery in selected subsets of SCC patients. For now, surgery remains the preferred option after nCRT when feasible, but non-operative strategies are increasingly viable alternatives in specific clinical contexts.

### 3.2. EGJ and Esophageal AC

#### 3.2.1. Pre- and Peri-Operative Approach

The management of resectable esophageal and EGJ AC has undergone significant transformation over the last two decades, largely shaped by a succession of randomized trials investigating perioperative chemotherapy, neoadjuvant chemoradiotherapy, and more recently, immune checkpoint inhibitors [23,25] [Table 2].

Initial landmark studies such as MAGIC and ACCORD07-FFCD 9703 demonstrated that perioperative chemotherapy improved OS compared to surgery alone [11,12]. The MAGIC trial reported a 5-year OS rate of 36% with ECF (epirubicin, cisplatin, and fluorouracil) versus 23% with surgery alone (HR 0.75; *p* = 0.009). In addition, perioperative chemotherapy significantly improved PFS, with 5-year PFS of 12.8% vs. 7.1% (HR 0.66; *p* < 0.001). Similarly, the ACCORD07-FFCD 9703 study confirmed the benefit of perioperative CF (cisplatin + 5-FU), reporting a 5-year OS of 38% vs. 24% (HR 0.69; *p* = 0.02) and a significant improvement in disease-free survival (DFS) (5-year DFS: 34% vs. 19%; HR 0.65; *p* = 0.003).

The FLOT4-AIO trial marked a major advance, establishing FLOT chemotherapy (5-FU, leucovorin, oxaliplatin, and docetaxel) as the new perioperative standard of care in resectable gastric and EGJ adenocarcinoma [10]. In this phase II/III trial including stage Ib–III patients, perioperative FLOT (4 pre- and 4 postoperative cycles) was compared to ECF/ECX (epirubicin, cisplatin, and 5-FU or capecitabine). FLOT significantly improved OS (mOS: 50 vs. 35 months; HR = 0.77; *p* = 0.012) and DFS (median DFS: 30 vs. 18 months; HR = 0.75; *p* = 0.0036). A higher pathological complete response (pCR) was also observed with FLOT (16% vs. 6%), supporting its adoption as the preferred regimen in the perioperative setting.

Parallel to chemotherapy development, nCRT also gained traction, notably through the CROSS trial [3,7]. While this trial included both AC and SCC, AC patients still showed a survival benefit with nCRT compared to surgery alone (10-year OS: 36% vs. 26%; *p* = 0.061) Importantly, nCRT was associated with a pCR rate of 23% in AC, lower than in SCC (49%), yet clinically meaningful.

To further clarify the optimal strategy in esophageal and EGJ AC, several head-to-head trials compared perioperative chemotherapy to nCRT. The Neo-AEGIS study, which included patients receiving either CROSS-based nCRT or perioperative chemotherapy (85% of patients receiving ECF/X), reported no significant difference in 3-year OS (55% vs. 57%; HR 1.03; *p* = 0.82), though pCR was higher with nCRT (HR 0.33; *p* = 0.012) [47]. The phase III ESOPEC trial compared perioperative FLOT chemotherapy with nCRT (CROSS) in resectable esophageal or junctional AC [48]. mOS was 66 months with FLOT versus 37 months with CROSS (HR = 0.70; *p* = 0.01). Notably, 3-year PFS was also improved with FLOT (51.6% vs. 35%; HR 0.66). Compared with nCRT, perioperative chemotherapy with FLOT improved survival through better systemic tumor control with a reduction in distant tumor recurrences (3-year cumulative incidences 31.5% v 47.2%, HR, 0.59; *p* = 0.002), while locoregional efficacy was similar [13]. These results validate FLOT as the preferred perioperative strategy for resectable esophageal and EGJ AC.

The phase III TOPGEAR trial evaluated whether adding preoperative chemoradiotherapy to perioperative chemotherapy could improve outcomes in resectable EGJ AC [14]. A total of 574 patients were randomized to receive perioperative ECF or FLOT alone, or the same regimen combined with preoperative chemoradiotherapy (45 Gy in 25 fractions with continuous fluorouracil infusion). While the addition of radiotherapy increased pathological complete response (17% vs. 8%), this did not translate into survival improvement. mOS was comparable between groups (46 vs. 49 months; HR = 1.05; *p*: NS), and no significant differences were observed in mPFS (31 vs. 32 months, HR: 0.66; *p*: NS) or 5-year PFS (40% in both arms). Notably, only 33% of patients in TOPGEAR actually received FLOT, raising similar concerns as in the Neo-AEGIS trial, which also suffered from suboptimal chemotherapy intensification. The phase III CRITICS study investigated whether adjuvant chemoradiotherapy could improve outcomes over chemotherapy alone following neoadjuvant chemotherapy and surgery in patients with stage IB–IVA gastric or EGJ AC [49]. A total of 788 patients were randomized after preoperative ECX and curative surgery to receive either postoperative chemotherapy or chemoradiotherapy (45 Gy in 25 fractions with concurrent cisplatin and capecitabine). No significant difference was observed in mOS (43 vs. 37 months; HR = 1.01; *p* = 0.90), nor in median event-free survival (mEFS) (28 vs. 25 months; HR = 0.99; *p* = 0.92). 5-year OS (42% vs. 40%) and EFS (39% vs. 38%) were also comparable. Collectively, these trials support the conclusion that radiotherapy offers no added benefit in either the adjuvant or neoadjuvant setting when combined with modern chemotherapy regimens.

The phase III CheckMate 577 trial evaluated adjuvant nivolumab versus placebo in 794 patients with residual disease after nCRT and R0 resection for esophageal or EGJ cancer [16,35]. The primary endpoint, DFS, was significantly improved with nivolumab (mDFS: 21.8 vs. 10.8 months; HR = 0.76; *p* < 0.001), and this benefit was sustained after 5 years of follow-up. OS, though not statistically significant in the overall population, favored nivolumab numerically (mOS: 51.7 vs. 35.3 months; HR = 0.85; *p* = 0.1064), with a more pronounced effect observed in key subgroups [35]. Notably, patients with PD-L1 CPS ≥ 1 experienced a significant OS improvement (45.5 vs. 33.5 months; HR = 0.79), while those with CPS <1 did not derive benefit (39.2 vs. 52.8 months; HR = 1.40). Similarly, benefit was greater in SCC (HR = 0.72) and esophageal tumors (HR = 0.69), whereas no advantage was observed in AC of the junction (HR = 1.14). Safety remained manageable and consistent with previous reports. These findings consolidate adjuvant nivolumab as a new standard of care for patients with residual disease following trimodal therapy, particularly in PD-L1–positive or SCC populations.

The phase III MATTERHORN trial investigated the addition of perioperative durvalumab (anti-PD-L1) to standard FLOT chemotherapy in patients with resectable gastric or EGJ AC [15]. A total of 948 patients were randomly assigned to receive either perioperative FLOT plus durvalumab or FLOT plus placebo, followed by surgery. The primary endpoint was EFS. At a median follow-up of 31.5 months, the 2-year EFS rate was significantly higher in the durvalumab group (67.4%) compared to placebo (58.5%), corresponding to a HR of 0.71 (*p* < 0.001). Subgroup analysis showed a consistent benefit regardless of PD-L1 expression, with an HR of 0.70 in patients with TAP ≥ 1%, and 0.77 in those with TAP < 1%. OS was also improved in the durvalumab arm (2-year OS: 75.7% vs. 70.4%; *p* = 0.03), although longer follow-up is needed to assess the durability of this effect. These results support the integration of durvalumab into perioperative treatment as a promising approach for localized gastric and EGJ AC. The phase III KEYNOTE-585 trial evaluated the addition of pembrolizumab to perioperative chemotherapy in patients with resectable stage II–IVA gastric or EGJ AC [18]. Despite a significant improvement in pCR (12.9% vs. 2.0%, *p* < 0.0001), the trial did not meet its primary endpoint, with 5-year EFS rates of 47% vs. 37% (HR = 0.81, *p* = NS). Subgroup analyses by PD-L1 status (CPS > 1 or < 1) showed no significant interaction. Similarly, OS was not significantly improved (mOS: 71.8 vs. 55.7 months; 5-year OS: 54% vs. 48%; HR = 0.86, *p* = NS). KEYNOTE-585 failed to demonstrate a significant EFS or OS benefit, likely due to multiple interim analyses and a suboptimal chemotherapy backbone, as only 20% of patients received FLOT. As seen in other trials lacking optimized regimens (e.g., TOPGEAR, Neo-AEGIS), the addition of immunotherapy without an effective perioperative chemotherapy backbone failed to translate into survival benefit. In contrast, MATTERHORN uniformly combined durvalumab with FLOT and showed a significant EFS gain, supporting both FLOT and immunotherapy (durvalumab) as essential components [50].

Taken together, current evidence supports FLOT as the optimal perioperative chemotherapy backbone for resectable esophageal and EGJ AC. Among immunotherapy-based strategies, the addition of perioperative durvalumab, as demonstrated in MATTERHORN, showed a statistically significant and consistent improvement in EFS and OS across PD-L1 subgroups, reinforcing its potential as a new standard of care irrespective of PD-L1 status. In contrast, pembrolizumab failed to improve survival in KEYNOTE-585, despite increasing pCR and EFS, likely due to a suboptimal chemotherapy backbone and limited FLOT use. nCRT followed by adjuvant nivolumab, validated in CheckMate 577, remains a valuable alternative in patients who are not candidates for upfront FLOT-based therapy. However, its benefit appears more restricted, with OS improvement mostly observed in patients with PD-L1 CPS ≥ 1, SCC histology, and esophageal primary tumors, while junctional AC derived no apparent advantage. Taken together, these data suggest that perioperative FLOT–durvalumab combination is now the recommended treatment option for esophageal and EGJ AC, irrespective of PD-L1 tumor status. This latter could help refine the benefits of adjuvant nivolumab treatment in patients treated with nCRT and surgery.

**Table 2 cancers-17-03603-t002:** Pre- and peri-operative key trials in Esophageal/EGJ AC.

Study	Population	Intervention Arm	Control Arm	Primary Endpoint	Outcomes
MAGIC (phase III) [11]	*n* = 503 Gastric/EGJ AC (100%) Stage II–III	Periop ECF + surgery	Surgery alone	OS	- 5-year OS: 36% vs. 23% (HR: 0.75, *p* = 0.009) - 5-year PFS: 12.8% vs. 7.1% (HR 0.66; *p* < 0.001)
FNCLCC ACCORD07-FFCD9703 (phase III) [12]	*n* = 224 Gastric/EGJ AC (100%) Stage I–IV	Periop CF + surgery	Surgery alone	OS	- 5-year OS: 38% vs. 24% (HR: 0.69; *p* = 0.02) - 5-year DFS: 34% vs. 19% (HR 0.65, *p* = 0.003)
FLOT4 (phase II/III) [10]	*n* = 716 Gastric/EGJ AC (100%) Stage Ib–III	Periop FLOT + surgery	ECF/ECX + surgery	OS	- mOS: 50 months vs. 35 months (HR: 0.77, *p* = 0.012) - mDFS: 30 months vs. 18 months (HR 0.75; *p* = 0.0036)
CROSS (phase III) [3,8]	*n* = 366 SCC (23%)-Gastric/EGJ AC (75%) Stage II–IIIa	nCRT (Carboplatin/paclitaxel) + RT 41.4 Gy in 23 daily fractions) + surgery	Surgery alone	OS	- mOS: 49.4 vs. 24.0 months (HR 0.657, *p* = 0.003) - 10-year OS (AC subgroup): 36% vs. 26% (*p* = 0.061) - R0 resection 92% vs. 69% (*p* < 0.001) - pCR (nCRT group): 49% for SCC vs. 23% for AC
Neo-AEGIS (phase III) [47]	*n* = 377 Gastric/EGJ AC (100%) Stage Ib–III	Periop ECF + surgery (MAGIC) Periop FLOT + surgery	nCRT: RT 41·4 Gy in 23 fractions + carboplatin/paclitaxel + surgery (CROSS)	OS	- mOS: 48.0 months vs. 49.2 months (HR 1.03; *p* = 0.82) - mDFS: 32.4 months vs. 24.0 months (HR 0.89; *p* = 0.41)
ESOPEC (phase III) [13]	*n* = 438 Gastric/EGJ AC (100%) Stage Ib–III	Periop FLOT + surgery	nCRT: RT 41·4 Gy in 23 fractions + carboplatin/paclitaxel (CROSS) + surgery	OS	- mOS: 66 months vs. 37 months (HR: 0.70; *p* = 0.01) - 3-year PFS: 51.6% vs. 35% (HR 0.66; *p*: NA)
TOPGEAR (phase II/III) [14]	*n* = 574 Gastric/EGJ AC (100%) Stage I–IV	Periop ECF or FLOT + preop CRT (45 Gy in 25 fractions + fluorouracile infusion) + surgery	Periop ECF or FLOT + surgery	OS	- mOS: 46 months vs. 49 months (HR: 1.05; *p*: NA) - mPFS: 31 months vs. 32 months (HR 0.98; *p*: NA) - pCR: 17% vs. 8%
CRITICS (phase III) [49]	*n* = 788 Gastric/EGJ AC (100%) Stage IB–IVA	Periop ECX + surgery	Preop ECX + surgery + postop CRT	OS	- mOS: 43 months vs. 37 months (HR: 1.01; *p* = 0.90) - mEFS: 28 months vs. 25 months (HR 0.99; *p* = 0.92)
CheckMate 577 (phase III) [16,35]	*n* = 794 SCC (29%)-Gastric/EGJ AC (71%) Stage II–III	nCRT + surgery + adjuvant Nivolumab	NCRT + surgery + adjuvant placebo	DFS	- mDFS: 21.8 months vs. 10.8 months (HR 0.76; *p* < 0.001) - mOS: 51.7 months vs. 35.3 months (HR: 0.85; *p* = 0.1064) If CPS ≥ 1: mOS: 45.5 months vs. 33.5 months (HR: 0.79; *p*: NA) If CPS < 1: mOS: 39.2 months vs. 52.8 months (HR 1.40; *p*: NA) AC subgroup: HR: 1.14
MATTERHORN (phase III) [15]	*n* = 948 Gastric/EGJ AC (100%) Stage I–IV (M0)	Periop FLOT + periop Durvalumab + surgery	Periop FLOT + surgery	EFS	- 2-year EFS: 67.4% vs. 58.5% (HR 0.71; *p* < 0.001) if TAP ≥ 1% HR 0.70 if TAP < 1%: HR 0.77 - 2-year OS: 75.7% vs. 70.4% (*p* = 0.03)
Keynote 585 (phase III) [18]	*n* = 804 Gastric/EGJ AC (100%) Stage II–IVa	Periop CT (Cisplatin + Fluorouracil or Capecitabine) + Pembrolizumab + surgery	Periop CT (Cisplatin + Fluorouracil or Capecitabine) + surgery	pCR, EFS, OS	- pCR: 12.9% vs. 2.0% (*p* < 0.0001) - mEFS: 44.4 months vs. 25.7 months (HR 0.81) Subgroup analyses (PD-L1 status): CPS > 1 (HR 0.86), CPS < 1 (HR 0.83) - mOS: 71.8 months vs. 55.7 months (HR 0.86)

Abbreviations: Adenocarcinoma (AC), Chemotherapy (CT), Chemoradiotherapy (CRT), Cisplatin + infused Fluorouracil (CF), Epirubicin + Cisplatin + Capecitabine (ECX), Epirubicin + Cisplatin + infused Fluorouracil (ECF), Esophagogastric Junction (EGJ), Event-Free Survival (EFS), Fluorouracil + Leucovorin + Oxaliplatin and Docetaxel (FLOT), Hazard Ratio (HR), median Event-Free Survival (mEFS), median Overall Survival (mOS), neoadjuvant Chemoradiotherapy (nCRT), Not available (NA), Overall Survival (OS), pathologic Complete Response (pCR), Perioperative (Periop), Postoperative (Postop), Preoperative (Preop), Programmed Death Ligand 1 (PD-L1), Radiotherapy (RT), Squamous Cell Carcinoma (SCC).

#### 3.2.2. Adjuvant Approach

The role of adjuvant therapy in localized esophageal and EGJ AC has gradually declined in Western practice due to the widespread adoption of neoadjuvant and perioperative strategies [10,35,51]. However, it remains a relevant approach in two specific contexts: in Eastern countries where upfront surgery followed by adjuvant chemotherapy is still commonly employed [52], and in Western settings when patients have not received neoadjuvant treatment, most often due to contraindications or emergent surgical indications [25,51]. In these latter cases, recommendations are largely based on the GASTRIC meta-analysis, which confirmed a significant survival benefit for fluorouracil-based adjuvant chemotherapy (over surgery alone) following curative resection of gastric and EGJ cancers (HR for OS: 0.82; *p* < 0.001) [53]. The key trials evaluating adjuvant strategies are summarized in Table 3.

In Asian populations, adjuvant chemotherapy following D2 surgery remains the standard of care, as demonstrated in pivotal phase III trials. In ACTS-GC, adjuvant S-1 monotherapy significantly improved 3-year OS compared to surgery alone (80.1% vs. 70.1%; HR = 0.68; *p* = 0.003), with a notable benefit also observed in 3-year relapse-free survival (RFS: 72.2% vs. 59.6%; HR = 0.62; *p* < 0.001) [54]. Similarly, the CLASSIC trial showed that adjuvant CAPOX significantly improved both DFS and OS. The 5-year DFS was 68% with CAPOX versus 53% with surgery alone (HR = 0.58; *p* < 0.0001), and the 5-year OS was 78% vs. 69% (HR = 0.66; *p* = 0.0015) [55]. These results firmly established fluoropyrimidine-based adjuvant chemotherapy as standard practice following curative surgery in this population.

The role of adjuvant CRT has been evaluated in several randomized trials with heterogeneous outcomes depending on surgical extent. The MacDonald trial randomized 556 patients with resected gastric/EGJ AC to surgery alone or surgery followed by adjuvant CRT (45 Gy with concurrent 5-FU and leucovorin) [56]. CRT significantly improved mOS (36 vs. 27 months; HR = 1.52; *p* = 0.005). However, only 10% of patients underwent D2 lymphadenectomy, suggesting that the benefit of CRT may be limited to cases with suboptimal lymph node dissection. In contrast, the phase III ARTIST and ARTIST-II trials investigated CRT following D2 surgery [57,58]. In ARTIST, adjuvant XP (capecitabin, cisplatin) was compared to XP plus CRT and failed to demonstrate any significant improvement in DFS or OS. ARTIST-II confirmed these findings by comparing S-1, SOX, and SOX plus CRT in node-positive patients, again without showing added benefit from radiotherapy. These data reinforce that in patients undergoing adequate D2 resection, CRT does not improve outcomes beyond those achieved with modern chemotherapy regimens alone.

The emergence of immunotherapy has brought renewed attention to the adjuvant setting. Contrary to the significant improvement observed in DFS, the CheckMate 577 trial did not demonstrate a statistically significant OS benefit with adjuvant nivolumab [16,35].

To improve outcomes in high-risk patients after perioperative FLOT, the phase II VESTIGE trial explored whether switching to adjuvant nivolumab plus ipilimumab (anti-CTLA-4) could be superior to continuing FLOT after surgery in patients with ypN+ disease and/or R1 resection [59]. The study did not meet its primary endpoint, with a significant detrimental median DFS of 11.4 months in the immunotherapy arm versus 20.8 months in the chemotherapy arm (HR = 1.55; *p* = 0.02). These results suggest that early switch to dual immunotherapies in patients with high risk for relapse may compromise outcomes compared to completing FLOT.

The retrospective SPACE-FLOT study involved 1887 patients evaluated the relevance of continuing adjuvant FLOT according to pathological response after neoadjuvant FLOT [60]. Patients were classified as complete responders (pCR), partial responders (PR), or minimal responders (MR). No survival benefit was seen with adjuvant FLOT in the pCR or MR subgroups, but a significant DFS (HR = 0.68; *p* < 0.001) and OS (HR = 0.55; *p* < 0.001) improvement was observed in the PR subgroup. These findings suggest that tailoring the adjuvant strategy based on pathological response may help identify patients most likely to benefit from continued chemotherapy. However, given the retrospective nature of this study and the results of the VESTIGE study, it is still recommended to continue adjuvant FLOT chemotherapy even in the absence of response and pathological risk factors.

The phase III ATTRACTION-5 trial evaluated the addition of adjuvant nivolumab to S-1 following D2 gastrectomy in 755 patients with stage IIIa–IIIc esophageal or EGJ AC [61]. The trial failed to meet its primary endpoint, with 3-year RFS rates of 68.4% in the nivolumab group versus 65.3% with placebo (HR = 0.90; *p* = 0.44), and no significant OS benefit (HR = 0.88). However, exploratory subgroup analyses suggested that patients with high-risk features, particularly pathological stage IIIc and PD-L1 tumor-cell expression ≥1%, may derive greater benefit from adjuvant nivolumab. These results imply a potential prognostic or predictive value of PD-L1 expression in this context, although the small sample size of subgroups warrants cautious interpretation.

In summary, adjuvant therapy remains a valid option in selected cases with esophageal and EGJ AC. While trials evaluating adjuvant immunotherapy have shown some improvement in DFS, none demonstrated a clear OS benefit. These heterogeneous outcomes underline the limited efficacy of late escalation strategies and reinforce the importance of improved upfront patient stratification. Integrating biomarkers such as PD-L1 status or pathological response as suggested in CheckMate 577, SPACE-FLOT and ATTRACTION-5 may help refine benefit of adjuvant treatment.

**Table 3 cancers-17-03603-t003:** Key adjuvant trials in Esophageal/EGJ AC.

Study	Population	Intervention Arm	Control Arm	Primary Endpoint	Outcomes
ACTS-GC (phase III) [54]	*n* = 1059 Gastric AC post-D2 surgery (R0 resection) (Asia) Stage II–IIIb	Surgery + adjuvant S-1 monotherapy	Surgery alone	OS	- 3-year OS: 80.1% vs. 70.1% (HR 0.68; *p* = 0.003) - 3-year RFS: 72.2% vs. 59.6% (HR 0.62; *p* < 0.001)
CLASSIC (phase III) [55]	*n* = 1035 Gastric AC post-D2 surgery (R0 resection) within 6 weeks before randomization (Asia) Stage II–IIIb	Surgery + adjuvant SOX	Surgery alone	DFS	- 5-year DFS: 68% vs. 53% (HR 0.58; *p* < 0.0001) - 5-year OS: 78% vs. 69% (HR 0.66; *p* = 0.0015)
MacDonald (phase III) [56]	*n* = 556 Resected gastric/EGJ AC (<10% D2 lymphadenectomy) Stage Ib–IV (M0)	Surgery + adjuvant CRT (RT 45 Gy + concurrent 5-FU and leucovorin)	Surgery alone	OS	- mOS: 36 months vs. 27 months (HR 1.35; *p* = 0.005) - mDFS: 30 months vs. 19 months (HR 1.52; *p* < 0.001)
ARTIST (phase III) [57]	*n* = 458 Gastric AC post-D2 surgery (R0 resection) Stage II–IV (M0)	Surgery + adjuvant CT (XP) + adjuvant CRT	Surgery + adjuvant CT (XP)	DFS	- 3-year DFS: 78.2% vs. 74.2% (*p* = 0.0862) - 5-year OS: 73% vs. 75% (HR 1.13; *p* = 0.527)
ARTIST II (phase III) [58]	*n* = 546 Gastric AC post-D2 surgery (R0 resection) Stage II–IV (M0)	Surgery + adjuvant SOX Surgery + adjuvant SOXRT	Surgery + adjuvant S-1	DFS	- 3-year DFS: 74.3% (SOX) vs. 64.8% (S-1) (HR 0.69; *p* = 0.042) - 3-year DFS: 72.8% (SOXRT) vs. 64.8% (S-1) (HR 0.72; *p* = 0.074)
CheckMate 577 (phase III) [16,35]	*n* = 794 Gastric/EGJ cancer (SCC 29%-AC 71%) post-nCRT & surgery (R0 resection) Stage II–III	nCRT + surgery + adjuvant Nivolumab	nCRT + surgery + adjuvant placebo	DFS	- mDFS: 21.8 months vs. 10.8 months (HR 0.76; *p* < 0.001) - mOS: 51.7 months vs. 35.3 months (HR: 0.85; *p* = 0.1064) If CPS ≥ 1: mOS: 45.5 months vs. 33.5 months (HR: 0.79, *p*: NA) If CPS < 1: mOS: 39.2 months vs. 52.8 months (HR 1.40, *p*: NA)
VESTIGE (phase II) [59]	*n* = 195 Gastric/EGJ AC post-preop FLOT + surgery in high-risk (ypN+ and/or R1)	Preop FLOT + surgery + adjuvant Nivolumab + Ipilimumab	Preop FLOT + surgery + Adjuvant CT (FLOT)	DFS	- mDFS: 11.4 months vs. 20.8 months (HR 1.55, *p* = 0.02) - mOS: 27.6 months vs. 38.0 months (HR 1,32, *p* = 0.235)
ATTRACTION-5 (phase III) [61]	*n* = 755 Gastric AC post-D2 surgery (Asia) Stage IIIa-c	Surgery + adjuvant S-1 + Nivolumab	Surgery + adjuvant S-1 + placebo	RFS	- 3-year RFS: 68.4% vs. 65.3% (HR: 0.90; *p* = 0.44) - HR for OS: 0.88 (*p*: NA)

Abbreviatons: Capecitabine + Cisplatin (XP), Capecitabine + Oxaliplatin (SOX), Capecitabine + Oxaliplatin + Chemoradiotherapy (SOXRT), Chemotherapy (CT), Chemoradiotherapy (CRT), Disease-Free Survival (DFS), Esophagogastric Junction (EGJ), Fluorouracil + Leucovorin + Oxaliplatin and Docetaxel (FLOT), Hazard ratio (HR), median Disease-Free Survival (mDFS), median Overall Survival (mOS), neoadjuvant Chemoradiotherapy (nCRT), Not available (NA), oral 5-Fluorouracil (S-1), Overall Survival (OS), Relapse-Free Survival (RFS).

## 4. Perspectives

### 4.1. Biomarker Driven for Selected Therapies

The increasing understanding of the molecular landscape of esophageal and EGJ cancers is progressively shaping treatment strategies, although its integration into routine clinical management of locally advanced disease remains limited. Among the biomarkers investigated, dMMR/MSI-H emerged as the most clinically actionable to date. Although dMMR/MSI-H tumors are rare (occurring in less than 10% of localized esophageal and gastric adenocarcinomas), they are associated with a favorable prognosis in resectable disease after surgery [62]. However, in the pre-FLOT era, these tumors appeared to derive no benefit or even potential harm from adjuvant or perioperative chemotherapy [63]. Whether the FLOT regimen overcomes this potential negative effect remains uncertain [64]. Importantly, dMMR/MSI-H tumors have consistently demonstrated high sensitivity to immune checkpoint inhibitors. Subgroup analyses from perioperative immunotherapy trials, including CheckMate 577, MATTERHORN, KEYNOTE-585, have reported impressive DFS benefits for dMMR/MSI-H populations receiving immunotherapy compared to controls, although the small sample size in each case limits the statistical power and prevents definitive conclusions [15,16,18,35]. Dedicated trials such as NEONIPIGA and INFINITY further support the potential of neoadjuvant immunotherapy in dMMR/MSI-H tumors [20,21] [Table 4]. In NEONIPIGA trial [21], evaluating neoadjuvant nivolumab and ipilimumab and adjuvant nivolumab in 32 patients with locally advanced resectable gastric and EGJ dMMR/MSI-H AC, 58.6% of patients achieved a pCR after dual immune checkpoint blockade. Similar results (60% of pCR) were observed in the INFINITY study (18 patients treated with tremelimumab (anti-CTLA-4) and durvalumab) [20]. Interestingly, this trial evaluated also the opportunity to use an immune-based chemo-free regimen as a definitive treatment option to enable non-operative management (NOM) and spare surgery to patients already cured by immunotherapy (cohort 2 of the study). For the 18 patients included in this cohort 2, the study demonstrated a high rate of clinical complete response (cCR, 76%) at the planned restaging assessment. During NOM, only one case of relapse was observed, consisting of a local regrowth that was successfully managed with salvage surgery, with no instances of distant progression. In addition, 12 patients were able to avoid surgery without compromising oncologic outcomes and with an acceptable safety profile (12-month gastrectomy-free survival was 64.2%). Conversely, patients who did not achieve cCR underwent salvage surgery and subsequently did not experience disease recurrence. Most recently, Cercek et al. provided compelling evidence that a subset of the 21 patients with localized dMMR/MSI-H gastro-esophageal tumors treated with 6 months anti-PD-1 dostarlimab can safely forego surgery following complete clinical response to neoadjuvant immunotherapy, supporting the paradigm of immunotherapy-based organ preservation in this molecular subgroup [65]. While larger confirmatory phase III trials are still awaited, these findings have already prompted an update of the National Comprehensive Cancer Network (NCCN) guidelines regarding neoadjuvant or perioperative immunotherapy [25]. As treatment decisions may be influenced, dMMR/MSI-H status should be assessed in all newly diagnosed cases non eligible for endoscopic resection to guide optimal therapeutic management.

HER2 (ERBB2) overexpression or amplification occurs in approximately 10–25% of gastric GEJ AC with the highest rates observed in tumors located at the EGJ [66]. Despite its established role in the metastatic setting, HER2-targeted strategies have not yet translated into practice-changing benefits in locally advanced disease. The following phase II and III trials have investigated HER2-targeted strategies in the perioperative or trimodality treatment of localized gastric and EGJ adenocarcinomas [Table 4]. The phase III RTOG-1010 trial evaluated trastuzumab added to trimodality therapy (chemoradiation followed by surgery) but failed to show significant improvement in DFS or OS [67]. The phase II PETRARCA trial evaluated the addition of trastuzumab ± pertuzumab to perioperative chemotherapy and showed higher pCR (12% vs. 35%; *p* = 0.02) and nodal negativity rates (39% vs. 68%) with the dual HER2 blockade, but at the cost of increased toxicity [68]. This trial was prematurely terminated before progressing to phase III due to negative outcomes and the absence of a survival benefit in the JACOB study, which evaluated chemotherapy plus trastuzumab and pertuzumab in metastatic HER2-positive gastric and EGJ adenocarcinoma [69] In the HER-FLOT phase II trial, the addition of trastuzumab to perioperative FLOT resulted in a pathological complete response (pCR) rate of 21.4%, without clear survival benefit [70]. The INNOVATION randomized phase II trial enrolled patients with HER2-positive gastric or EGJ AC and assigned them to perioperative chemotherapy alone (*n* = 35), chemotherapy plus trastuzumab (*n* = 67), or chemotherapy plus trastuzumab and pertuzumab (*n* = 70) [19]. The initial regimen of cisplatin plus capecitabine was later replaced by FLOT after it became the new standard of care. At five years, PFS did not differ significantly among the groups (51.9% vs. 61.0% vs. 47.9%), nor did OS (60.5% vs. 67.5% vs. 62.6%). Based on these findings, FLOT-Durvalumab without HER-2 targeted therapy remains the standard-of-care treatment for patients with locally advanced, HER2-positive gastroesophageal adenocarcinoma.

In a large ctDNA/tissue Next Generation Sequencing (NGS) study, patients with HER2 or EGFR amplification identified by ctDNA and/or tissue NGS derived significant survival benefit from matched targeted therapies, with mOS of 26.3 vs. 7.4 months for HER2 (*p* = 0.002) and 21.1 vs. 14.4 months for EGFR (*p* = 0.01) [22]. This aligns with earlier findings from Leal et al., where postoperative ctDNA detection was associated with shorter OS (HR 3.3; *p* = 0.28) [71]. These findings support the idea that ctDNA may help identify patients eligible for treatment intensification or, conversely, de-escalation, although randomized trials are still lacking to validate this approach.

PD-L1 expression can be assessed using different scoring systems: the Combined Positive Score (CPS), which accounts for PD-L1 staining on both tumor and immune cells (cut-offs usually ≥1, ≥5 or ≥10); the Tumor Proportion Score (TPS), which measures the percentage of PD-L1–positive tumor cells only; and the Tumor Area Positivity (TAP), reflecting the proportion of the tumor area occupied by PD-L1–positive cells, mainly used in Asian trials. Although PD-L1 has not consistently emerged as a predictive biomarker in the curative setting based on randomized trials, it is now recommended by NCCN guidelines in localized adenocarcinomas to consider the addition of immunotherapy to perioperative chemotherapy, highlighting its increasing relevance in clinical practice [25]. In all major perioperative or adjuvant immunotherapy trials CheckMate 577, MATTERHORN, KEYNOTE585, and DANTE PD-L1 expression was used as a stratification factor rather than a selection criterion. Subgroup analyses from CheckMate 577 suggested a greater DFS benefit from adjuvant nivolumab in patients with CPS ≥ 1, though an absolute DFS gain was also observed in CPS < 1 patients [15,34]. However, updated OS data from this trial showed no significant survival advantage in the overall population, and subgroup analyses according to PD-L1 CPS suggested heterogeneous outcomes. Patients with CPS ≥ 1 appeared to derive a numerical benefit from adjuvant nivolumab, with a mOS of 45.5 versus 33.5 months (HR 0.79), whereas those with CPS < 1 had an opposite trend, with mOS of 39.2 versus 52.8 months (HR 1.40), raising concerns regarding the consistency of benefit across biomarker-defined subgroups [35]. Similarly, exploratory analyses from MATTERHORN and KEYNOTE-585 suggested a potential association between PD-L1 expression and pathological complete response, with pCR rates of 19.2% vs. 14.1% in patients with TAP ≥ 1% vs. <1% in MATTERHORN, and 17.2% vs. 11.3% in KEYNOTE585 (CPS ≥ 1 vs. <1). However, this did not translate into a clear correlation with EFS or OS, as both studies showed consistent benefit across PD-L1 subgroups, limiting the predictive value of PD-L1 status in this setting [15,18,35].

Emerging biomarkers such as claudin 18.2 and FGFR2b have garnered increasing interest, particularly in advanced and metastatic gastric and EGJ AC, where targeted therapies like zolbetuximab and bemarituzumab have demonstrated meaningful clinical benefit in phase II/III trials [72,73]. Despite their potential, their relevance in the curative setting remains undefined, with no perioperative trials yet establishing efficacy. Moreover, novel therapeutic platforms such as CAR-T cells targeting claudin 18.2 or other gastrointestinal-specific antigens are being actively explored in metastatic disease, showing early signs of safety and activity [74]. Translating these innovations to localized disease will require careful evaluation through biomarker-driven perioperative studies, which are critical to advancing personalized therapeutic strategies.

**Table 4 cancers-17-03603-t004:** Trials with biomarker selected therapy in locally advanced esophageal and EGJ AC.

Study	Population	Intervention Arm	Control Arm	Primary Endpoint	Outcomes
NEONIPIGA (phase II) [21]	*n* = 32 (*n* = 1 metastatic, wrongly included and excluded of analysis) Resectable dMMR/MSI-H Gastric/EGJ AC Stage Ib–III	Preop Nivolumab + Ipilimumab + Surgery + adjuvant Nivolumab	No control arm	pCR	- pCR: 58.6%
INFINITY (phase II) [20]	*n* = 15 in cohort 1 *n* = 18 in cohort 2 Resectable dMMR/MSI-H Gastric/EGJ AC Stage Ib–III	Cohort 1: Preop Tremelimumab + Durvalumab + mandatory surgery Cohort 2: Preop Tremelimumab + Durvalumab + Surgery only if no cCR (otherwise NOM with surveillance)	No control arm	Cohort 1: pCR Cohort 2: cCR	Cohort 1: pCR: 60% - 2-year PFS: 66.7%. 2-year OS: 73.3% Cohort 2: cCR 76% - 1-year PFS: 93.8%. 1-year OS: 100% - 1-year Gastrectomy-Free Survival: 64.2%
RTOG1010 (phase III) [67]	*n* = 203 HER2-positive Gastric/EGJ AC Stage Ib–IIIb	Preop CRT + Trastuzumab + surgery	Preop CRT + surgery	DFS	- mDFS: 19.6 months vs. 14.2 months (HR 0.99; *p* = 0.97) - mOS: 38.5 months vs. 38.9 months (HR 1.04; *p* = 0.85)
PETRARCA (phase II) [68]	*n* = 81 HER2-positive Gastric/EGJ AC Stage Ib–III	Periop FLOT + Trastuzumab + Pertuzumab + surgery	Periop FLOT + surgery	pCR	- pCR 35% vs. 12% (*p* = 0.019) - 2-year DFS: 70% vs. 54% (HR 0.58; *p* = 0.130) - 2-year OS: 84% vs. 77% (HR 0.56; *p* = 0.228)
HER-FLOT (phase II) [70]	*n* = 56 HER2-positive Gastric/EGJ AC Stage Ib–III	Periop Trastuzumab + FLOT + Surgery	No control arm	PCR (>20%)	- pCR 21.4% - R0 resection: 92.9% - mDFS 42.5 months - 3-year OS: 82.1%
INNOVATION (phase II) [19]	N = 172 HER2-positive Gastric/EGJ AC Stage Ib–III	Periop CT (FLOT, with FOLFOX or SOX as alternative) + Trastuzumab + surgery (Arm B) Periop CT (FLOT, with FOLFOX or SOX as alternative) + Trastuzumab + Pertuzumab + surgery (Arm C)	Periop CT (FLOT, with FOLFOX or SOX as alternative) (Arm A)	mpRR	mpRR: 33% (arm A) vs. 53.3% (arm B) vs. 37.9% (arm C) after amending the protocol mpRR: 8.3% (arm A) vs. 16.7% (arm B) vs. 12.5% (arm C) before amending the protocol - 5-year PFS: 51.9% (arm A) vs. 61.0% (arm B) (HR 0.88) - 5-year PFS: 51.9% (arm A) vs. 47.9% (arm C) (HR 1.40) - 5-year OS: 60.5% (arm A) vs. 67.5% (arm B) (HR 0.89) - 5-year OS: 60.5% (arm A) vs. 62.6% (arm C) (HR 1.29)

Abbreviations: Adenocarcinoma (AC), Capecitabine + Oxaliplatin (SOX), Chemotherapy (CT), Chemoradiotherapy (CRT), clinical Complete Response (cCR), Disease-Free Survival (DFS), Esophagogastric Junction (EGJ), Fluorouracil + Leucovorin + Oxaliplatin (FOLFOX), Fluorouracil + Leucovorin + Oxaliplatin and Docetaxel (FLOT), Hazard Ratio (HR), Human Epidermal Growth Factor Receptor 2 (HER2), major pathological Response Rates (mpRR), median Disease-Free Survival (mDFS), median Overall Survival (mOS), median Progression-Free Survival (mPFS), Mismatch Repair Deficiency or Microsatellite Instability-High (dMMR/MSI-H), Non-Operative Management (NOM), Overall Survival (OS), pathologic Complete Response (pCR), Perioperative (Periop), Preoperative (Preop), Progression-Free Survival (PFS).

### 4.2. Assessing the Response to Better Define Treatment Sequence and Modalities

Accurate assessment of treatment response is a cornerstone of precision oncology and is particularly relevant in the evolving landscape of multimodal therapy for locally advanced esophageal and EGJ cancers. Conventional evaluation involving imagery and endoscopy, remain insufficiently sensitive to detect residual microscopic disease or predict long-term outcomes, particularly after neoadjuvant chemoradiotherapy or perioperative chemotherapy ± immunotherapy. Recent data from the SANO and INFINITY trials also underscore growing interest in non-operative management for complete responders, although this approach is not yet standard of care [20,39]. Consequently, novel tools such as ctDNA and functional imaging are being explored as complementary modalities to guide clinical decisions [75].

CtDNA has emerged as a promising biomarker for risk stratification, monitoring therapeutic efficacy, and early detection of relapse in locally advanced gastroesophageal cancers. Preliminary findings suggest that ctDNA status before treatment may outperform PET-assessed nodal status in predicting DFS [76]. Furthermore, ctDNA clearance following neoadjuvant therapy has been correlated with better treatment outcomes. In other observational data, the presence of ctDNA prior to surgery was linked to a higher likelihood of recurrence [77]. Some exploratory studies have even investigated ctDNA as a tool for response-adapted strategies, such as surveillance in patients with favorable ctDNA kinetics during chemoradiotherapy [78]. Despite encouraging signals, these applications remain investigational, and larger prospective trials are needed to validate ctDNA’s utility in routine clinical decision-making.

Another investigational strategy is functional imaging, notably 18F-FDG PET/CT, which provides early metabolic response data. In the MUNICON-I trial, metabolic response was assessed as early as day 14 of neoadjuvant chemotherapy, allowing for early treatment adaptation. Early metabolic responders had a significantly improved mOS (not reached) compared to 25.8 months in non-responders (HR = 2.13; *p* = 0.015). However, the follow-up MUNICON-II study failed to show a survival benefit from intensifying treatment in non-responders, thereby limiting the applicability of PET-guided strategies in routine practice despite ongoing research interest [79].

Additional advanced imaging modalities, including diffusion-weighted magnetic resonance imaging or novel PET tracers (e.g., (Fibroblast Activation Protein Inhibitor (FAPI), Fluorothymidine (FLT)), are being explored but remain investigational [80].

Endoscopic and histopathologic reassessment strategies are being evaluated, particularly in organ-preserving approaches. The preSANO and SANO studies combined endoscopy, bite-on-bite biopsies, and EUS-FNA to detect residual disease after nCRT [8,44]. While sensitivity for detecting residual tumor remained moderate (60–75%), the SANO trial demonstrated that when integrated with clinical and radiologic assessment, this multimodal approach achieved sufficient negative predictive value to safely guide active surveillance in selected patients, with comparable OS to immediate surgery.

Altogether, integrating ctDNA profiling, functional imaging, and targeted biopsies could allow in the future real-time adaptation of therapeutic strategies. However, none of these tools are yet validated for treatment adaptation in routine clinical practice and should be restricted to controlled clinical trials or high-volume centers.

## 5. Conclusions

Over the past two decades, the management of localized esophageal and EGJ cancers has evolved significantly, guided by successive landmark trials. nCRT or dCRT remains the standard for SCC in Western practice. For AC, radiotherapy is no longer the preferred approach, as randomized trials comparing perioperative chemotherapy backbones with added chemoradiotherapy have shown no consistent survival benefit. However, CRT may still be appropriate in selected cases, such as borderline resectable tumors, in specific centers, or when surgical timing and logistics favor a CROSS-like approach. Perioperative FLOT chemotherapy, now combined with durvalumab, has emerged as the new standard of care based on the most recent evidence. These paradigm shifts underline the importance of histological subtyping in guiding curative strategies.

Among biomarkers, MSI stands out as the only robust predictor of benefit from immunotherapy in localized disease. MSI-high tumors are highly sensitive to immune checkpoint blockade, underscoring the clinical relevance of systematic MSI testing in this setting.

Moreover, evolving paradigms in response-adapted therapy such as PET-guided modifications or active surveillance protocols without surgery in case of complete response highlight a shift toward personalized treatment intensity. The integration of ctDNA as a potential marker as a potential marker of minimal residual disease may further enhance decision-making and allow dynamic adaptation of therapeutic strategies soon. In the future, trials integrating predictive biomarkers, tools for assessing complete response, and personalized strategies based on molecular tumor profiling will likely be key to curing patients while minimizing unnecessary treatment-related morbidity.

## Data Availability

No new data were created or analyzed in this study. Data sharing is not applicable to this article.

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
