# Peer review of "Current Management of Locally Advanced Esophageal and Esophagogastric Junction Cancers: Clinical Evidence and Evolving Strategies"

_cancers, 2025, doi:10.3390/cancers17223603_

Round 1
Reviewer 1 Report
Comments and Suggestions for Authors
Thank you for the opportunity to review this concise and well written review.
I have a few recommendations:
- Sometimes you use esophageal (l.120) /oesophageal (l.104) or esophagectomy / oesophagectomy (l. 93) -> decide on one and use throughout the whole manuscript
- Flowchart or other means of visiualisation would enhance understandability of the manuscript and add to its value
- Few minor grammar faults (l. 116 -> both histology)
- Tables (esp. Tables 1 and 3) are a bit confusing – try to unify outcomes, population etc.
- L.55 -> newer data for incidence and prevalence are available. Even in china SCC accounts for less than 90% now (doi 10.1016/j.jncc.2023.01.002).
- You switch between CPS, TPS, and TAP without definitions. Consider providing a short box explaining each score), scoring, and cut-offs.
- You conclude RT adds “no benefit” with modern chemo. Consider softening to e.g. “no consistent survival benefit in randomized trials comparing peri-op chemo backbones vs added RT,” as there are still cases in which CRT remains appropriate (borderline resectable, specific centers, or when surgery timing/logistics favour CROSS).
Author Response
Comment 1 : Sometimes you use esophageal (l.120) /oesophageal (l.104) or esophagectomy / oesophagectomy (l. 93) -> decide on one and use throughout the whole manuscript
Response 1 : We thank the reviewer for pointing this out. We have revised the manuscript to ensure consistent use of the terminology (esophagectomy, l.94; esophageal, l.105).
Comment 2 : Flowchart or other means of visiualisation would enhance understandability of the manuscript and add to its value
Response 2 : We thank the reviewer for his suggestion. We preferred to keep the structure of the manuscript as it is, but we have clarified certain tables to make them clearer for the reader.
Comment 3 : Few minor grammar faults (l. 116 -> both histology)
Response 3 : We appreciate this correction. The text has been revised accordingly (both histologies, l.117).
Comment 4 : Tables (esp. Tables 1 and 3) are a bit confusing – try to unify outcomes, population etc.
Response 4 : We thank the reviewer for this valuable suggestion. Several studies included in these tables enrolled patients with different histological types, tumor locations, or PD-L1 expression levels. However, detailed results are not available for all of these subgroups.
Following your comment, we attempted to unify these subgroups by duplicating certain tables. However, this approach made the tables less readable and introduced considerable redundancy. For the sake of clarity and simplicity, we therefore decided not to separate the subgroups and to present the studies as they were originally reported.
We nevertheless retained in the tables the subgroup results that we considered particularly relevant to report. We hope that this presentation will meet your expectations. Alternatively, we could keep only the overall results in the tables and discuss the subgroup analyses within the main text of the manuscript.
We remain open to any further suggestions.
Comment 5 : L.55 -> newer data for incidence and prevalence are available. Even in china SCC accounts for less than 90% now (doi 10.1016/j.jncc.2023.01.002).
Response 5 : We agree with this comment and have updated the text to incorporate the recent Chinese registry data, emphasizing that SCC now accounts for less than 90% of cases (l.55).
Comment 6 : You switch between CPS, TPS, and TAP without definitions. Consider providing a short box explaining each score), scoring, and cut-offs.
Response 6 : We agree with this comment and have added concise definitions of CPS, TPS, and TAP in the manuscript for clarity (l.501).
Comment 7 : You conclude RT adds “no benefit” with modern chemo. Consider softening to e.g. “no consistent survival benefit in randomized trials comparing peri-op chemo backbones vs added RT,” as there are still cases in which CRT remains appropriate (borderline resectable, specific centers, or when surgery timing/logistics favour CROSS).
Response 7 : We agree with this comment and have modified the conclusion accordingly, softening the statement to reflect the lack of consistent survival benefit in randomized trials, while acknowledging that CRT may still be appropriate in selected situations (l.590).
Reviewer 2 Report
Comments and Suggestions for Authors
Authors present a fantastic review of the current data regarding neoadjuvant management of locally advanced distal esophageal and Ge junction cancer. The review is well written and very rigorous.
The only comment I have is line 504: Although is correct that PDL-1 has been used only fro post hoc analysis of trials, the current NCCN guidelines recommend PDL-1 evaluation of all adenocarcinomas to consider the addition of immunotherapy to FLOT therefore can be considered a biomarker currently used on clinical practice
Author Response
Comment 1 : Although is correct that PDL-1 has been used only fro post hoc analysis of trials, the current NCCN guidelines recommend PDL-1 evaluation of all adenocarcinomas to consider the addition of immunotherapy to FLOT therefore can be considered a biomarker currently used on clinical practice
Response 1 : We thank the reviewer for this valuable comment. We agree that, while PD-L1 expression has mainly been explored in post hoc analyses of clinical trials, current NCCN guidelines (v4.2025) already recommend the universal testing for PD-L1 by IHC for all newly diagnosed patients with esophageal and EGJ cancers who are candidates for treatment with PD-1 or PD-L1 inhibitors. We have revised the text accordingly to better reflect its emerging role in clinical practice (l.506).
Reviewer 3 Report
Comments and Suggestions for Authors
Multimodal treatment of locally advanced GC, based on chemo- radio- and immuno-therapy, is a complex therapeutic strategy nowadays, based on the biological characteristics of the tumor, its stage, the patient's conditions, and the accuracy of the clinical and instrumental assessment, as well. Despite the huge progress in technology and farmacology, many ongoing trials fail to demonstrate the definitive excellence of one treatment over another. Also the different experience in different Countries contributes to this uncertainty. An incomplete explanation can be the enormous eterogeneity of gastric cancer envirnment, both in the same tumor, and in different fases of its growth that contributes to its relative resistance to treatment.
The authors made a complete review of the newest literature about the treatment of esophageal squamous cell carcinoma and EJ adenocarcinoma. Since the observations above, I think a revision should be done, in order to enhance the quality of the article: Squamous cell carcinoma and adenocarcinoma of the esophagus are two different diseases; I think they should be treated separately; in the article, I often observe a kind of overlapping in many concepts, that can be a bit confused.
Definitely, I'd advice the authors to drop the cases of SCC and focus theyr review on the treatment of the adenocarcinoma alone.
Author Response
Comment 1 : Multimodal treatment of locally advanced GC, based on chemo- radio- and immuno-therapy, is a complex therapeutic strategy nowadays, based on the biological characteristics of the tumor, its stage, the patient's conditions, and the accuracy of the clinical and instrumental assessment, as well. Despite the huge progress in technology and farmacology, many ongoing trials fail to demonstrate the definitive excellence of one treatment over another. Also the different experience in different Countries contributes to this uncertainty. An incomplete explanation can be the enormous eterogeneity of gastric cancer envirnment, both in the same tumor, and in different fases of its growth that contributes to its relative resistance to treatment.
The authors made a complete review of the newest literature about the treatment of esophageal squamous cell carcinoma and EJ adenocarcinoma. Since the observations above, I think a revision should be done, in order to enhance the quality of the article: Squamous cell carcinoma and adenocarcinoma of the esophagus are two different diseases; I think they should be treated separately; in the article, I often observe a kind of overlapping in many concepts, that can be a bit confused.
Definitely, I'd advice the authors to drop the cases of SCC and focus theyr review on the treatment of the adenocarcinoma alone.
Response 1 : We sincerely thank the reviewer for this thoughtful comment. We fully agree that esophageal squamous cell carcinoma (SCC) and adenocarcinoma (AC) represent two distinct entities with different biological backgrounds, epidemiology, and therapeutic strategies. However, the aim of our review was precisely to provide a comprehensive overview comparing current and emerging treatment approaches across histologies, highlighting both shared principles and histology-specific considerations. Recent high-quality reviews have already focused exclusively on adenocarcinoma, and we felt it was important to offer a broader perspective encompassing both histologies, especially given the increasing availability of randomized trials reporting separate outcomes for SCC and AC.
Reviewer 4 Report
Comments and Suggestions for Authors
Thank you very much for giving me a good opportunity to review your article. This review provides an up-to-date synthesis of current and emerging treatment strategies for locally advanced esophageal and EGJ cancers. It is a meaningful review and this manuscript is well written. So I determined that this manuscript is worthy of acceptance to "Cancers" in its present form in the following respects. The following are the shortcomings that need to be corrected.
Minor comments:
- There are some spell or descriptive errors in text.
For example,
- “Conversely, HER2-targeted strategies,…” → “Conversely, human epidermal growth factor receptor 2 (HER2)-targeted strategies,…” (Page 1, Line 32-33)
- “circulating tumor DNA and functional imaging as….” → “circulating tumor deoxyribo nucleic acid (DNA) and functional imaging as….” (Page 1, Line 34-35)
- “dMMR/MSI-H, Epstein-Barr virus….” → “deficient mismatch repair (dMMR)/ Microsatellite Instability-High (MSI-H), Epstein-Barr virus….” (Page 2, Line 62)
- “Furthermore, the advent of liquid biopsies (ctDNA) and advanced imaging could enable real-time response adaptation [22].” → “Furthermore, the advent of liquid biopsies such as circulating tumor DNA (ctDNA) and advanced imaging could enable real-time response adaptation [22].” (Page 2, Line 73-74)
- “The role of HER-2 (for AC) and PD-L1 biomarkers….” → “The role of HER2 (for AC) and programmed cell death-ligand1 (PD-L1) biomarkers….” (Page 3, Line 115-116)
- “according to the American Joint Committee on Cancer AJCC/UICC TNM (tumor- node-metastasis) 8th edition staging system….” → “according to the American Joint Committee on Cancer (AJCC)/Union for International Cancer Control (UICC) tumor- node-metastasis (TNM) 8th edition staging system….” (Page 3, Line 107-108)
- “(OS: 49.4 vs. 24.0 months; HR = 0.66, p = 0.003)….” → “[overall survival (OS): 49.4 vs. 24.0 months; hazard ratio (HR) = 0.66, p = 0.003]….” (Page 3, Line 126-127)
- "In contrast, the FFCD 9901 trial enrolling patients…." → "In contrast, the Francophone de Cancérologie Digestive (FFCD) 9901 trial enrolling patients…." (Page 3, Line 129-130)
- "The Asian JCOG9907 trial compared neoadjuvant chemotherapy…." → "The Asian Japan Clinical Oncology Group (JCOG)9907 trial compared neoadjuvant chemotherapy…." (Page 4, Line 133)
- "3-year (progression-free survival) PFS did not…" → "3-year progression-free survival (PFS) did not…." (Page 4, Line 136-137)
- "The japanese JCOG1109 (NExT) trial…." → "The Japanese JCOG1109 (NExT) trial…." (Page 4, Line 140)
- "a significant benefit in (disease-free survival) DFS with…." → "a significant benefit in disease-free survival (DFS) with…." (Page 4, Line 152)
- "favored nivolumab numerically (mOS : 51.7 vs. ….)" → "favored nivolumab numerically [median OS (mOS) : 51.7 vs. ….]" (Page 4, Line 155-156)
- "Notably, patients with PD-L1 CPS ≥1 experienced…." → "Notably, patients with PD-L1 combined positive score (CPS) ≥1 experienced…." (Page 4, Line 157)
- "pivotal studies such as RTOG 166 85-01,…" → "pivotal studies such as Radiation Therapy Oncology Group (RTOG) 85-01,… " (Page 4, Line 166-167)
- "in patients with complete clinical response (cCR)…." → " in patients with clinical complete response (cCR)…. " (Page 5, Line 197-198)
- "nCRT (Fluor-ouracile/Cispla-tin) + RT 45 Gy : 25 fractions over 5 weeks + sur-gery" → "nCRT (Fluor-ouracil/Cispla-tin) + RT 45 Gy : 25 fractions over 5 weeks + sur-gery" (Page 6(2?), in Table 1)
- "RT 50Gy in 25 fractions over 5 weeks + CT (Cis-platin/fluor-ouracile)" → "RT 50Gy in 25 fractions over 5 weeks + CT (Cis-platin/fluor-ouracil)" (Page 7(3?), in Table 1)
- "(HR), High Dose (HD) local Progression-Free Survival (LPFS) median Disease-Free Survival (mDFS)…." → "(HR), High Dose (HD), local Progression-Free Survival (LPFS), median Disease-Free Survival (mDFS),…." (Page 7(3?), in Table 1 footnote)
- "chemotherapy significantly improved progression-free survival (PFS),…" → "chemotherapy significantly improved PFS,…" (Page 8(4?), Line 231) Because this is not the first appearance.
- "a significant improvement in DFS (5-year…." → "a significant improvement in disease-free survival (DFS) (5-year…." (Page 8(4?), Line 234) Because this is the first appearance in the text.
- "FLOT significantly improved overall survival (mOS: …." → "FLOT significantly improved OS (mOS: …." (Page 8(4?), Line 240-241) Because this is not the first appearance.
- "DFS (mDFS: 30 vs. 18…." → "DFS (median DFS: 30 vs. 18…." (Page 8(4?), Line 241)
- "A higher pCR was also observed…." → "A higher pathological complete response (pCR) rate was also observed…." (Page 8(4?), Line 242)
- "in resectable esophageal or junctional adenocarcinoma [48]. " → "in resectable esophageal or junctional AC [48]." (Page 8(4?), Line 255) Because this is not the first appearance.
- "nor in mEFS (median event-free 278 survival: 28 vs. 25 months; HR = 0.99; p = 0.92)." → "nor in median event-free survival (mEFS): 28 vs. 25 months; HR = 0.99; p = 0.92)." (Page 9(5?), Line 278-279)
- "using the CPS or TAP score, has not yet…." → "using the CPS or tumor area positivity (TAP) score, has not yet…." (Page 16(2?), Line 504)
- "a greater disease-free survival (DFS) benefit…." → "a greater DFS benefit…." (Page 16(2?), Line5095) Because this is not the first appearance.
- "Fluorouracile + Leucovorin + Oxaliplatin (FOLFOX),…." → "Fluorouracil + Leucovorin + Oxaliplatin (FOLFOX), …." (Page 18(4?), Line 537, in Table 4 footnote)
- "Circulating tumor DNA (ctDNA) has emerged…." → "CtDNA has emerged…." (Page 19(3?), Line 555)
- "including diffusion-weighted MRI or…." → "including diffusion-weighted magnetic resonance imaging (MRI) or…." (Page 19(3?), Line 574) Or you should not abbreviate due to the single use in manuscript.
- "The integration of circu-601 lating tumor DNA (ctDNA) as a potential marker…." → "The integration of ctDNA as a potential marker…." (Page 19(3?), Line 601-602)
Author Response
Comment 1 : There are some spell or descriptive errors in text.
- Conversely, HER2-targeted strategies,…” → “Conversely, human epidermal growth factor receptor 2 (HER2)-targeted strategies,…” (Page 1, Line 32-33)
- Circulating tumor DNA and functional imaging as….” → “circulating tumor deoxyribo nucleic acid (DNA) and functional imaging as….” (Page 1, Line 34-35)
- “dMMR/MSI-H, Epstein-Barr virus….” → “deficient mismatch repair (dMMR)/ Microsatellite Instability-High (MSI-H), Epstein-Barr virus….” (Page 2, Line 62)
- “Furthermore, the advent of liquid biopsies (ctDNA) and advanced imaging could enable real-time response adaptation [22].” → “Furthermore, the advent of liquid biopsies such as circulating tumor DNA (ctDNA) and advanced imaging could enable real-time response adaptation [22].” (Page 2, Line 73-74)
- “The role of HER-2 (for AC) and PD-L1 biomarkers….” → “The role of HER2 (for AC) and programmed cell death-ligand1 (PD-L1) biomarkers….” (Page 3, Line 115-116)
- “according to the American Joint Committee on Cancer AJCC/UICC TNM (tumor- node-metastasis) 8th edition staging system….” → “according to the American Joint Committee on Cancer (AJCC)/Union for International Cancer Control (UICC) tumor- node-metastasis (TNM) 8th edition staging system….” (Page 3, Line 107-108)
- “(OS: 49.4 vs. 24.0 months; HR = 0.66, p = 0.003)….” → “[overall survival (OS): 49.4 vs. 24.0 months; hazard ratio (HR) = 0.66, p = 0.003]….” (Page 3, Line 126-127)
- "In contrast, the FFCD 9901 trial enrolling patients…." → "In contrast, the Francophone de Cancérologie Digestive (FFCD) 9901 trial enrolling patients…." (Page 3, Line 129-130)
- "The Asian JCOG9907 trial compared neoadjuvant chemotherapy…." → "The Asian Japan Clinical Oncology Group (JCOG)9907 trial compared neoadjuvant chemotherapy…." (Page 4, Line 133)
- "3-year (progression-free survival) PFS did not…" → "3-year progression-free survival (PFS) did not…." (Page 4, Line 136-137)
- "The japanese JCOG1109 (NExT) trial…." → "The Japanese JCOG1109 (NExT) trial…." (Page 4, Line 140)
- "a significant benefit in (disease-free survival) DFS with…." → "a significant benefit in disease-free survival (DFS) with…." (Page 4, Line 152)
- "favored nivolumab numerically (mOS : 51.7 vs. ….)" → "favored nivolumab numerically [median OS (mOS) : 51.7 vs. ….]" (Page 4, Line 155-156)
- "Notably, patients with PD-L1 CPS ≥1 experienced…." → "Notably, patients with PD-L1 combined positive score (CPS) ≥1 experienced…." (Page 4, Line 157)
- "pivotal studies such as RTOG 166 85-01,…" → "pivotal studies such as Radiation Therapy Oncology Group (RTOG) 85-01,… " (Page 4, Line 166-167)
- "in patients with complete clinical response (cCR)…." → " in patients with clinical complete response (cCR)…. " (Page 5, Line 197-198)
- "nCRT (Fluor-ouracile/Cispla-tin) + RT 45 Gy : 25 fractions over 5 weeks + sur-gery" → "nCRT (Fluor-ouracil/Cispla-tin) + RT 45 Gy : 25 fractions over 5 weeks + sur-gery" (Page 6(2?), in Table 1)
"RT 50Gy in 25 fractions over 5 weeks + CT (Cis-platin/fluor-ouracile)" → "RT 50Gy in 25 fractions over 5 weeks + CT (Cis-platin/fluor-ouracil)" (Page 7(3?), in Table 1)
- "(HR), High Dose (HD) local Progression-Free Survival (LPFS) median Disease-Free Survival (mDFS)…." → "(HR), High Dose (HD), local Progression-Free Survival (LPFS), median Disease-Free Survival (mDFS),…." (Page 7(3?), in Table 1 footnote)
- "chemotherapy significantly improved progression-free survival (PFS),…" → "chemotherapy significantly improved PFS,…" (Page 8(4?), Line 231) Because this is not the first appearance.
- "a significant improvement in DFS (5-year…." → "a significant improvement in disease-free survival (DFS) (5-year…." (Page 8(4?), Line 234) Because this is the first appearance in the text.
- "FLOT significantly improved overall survival (mOS: …." → "FLOT significantly improved OS (mOS: …." (Page 8(4?), Line 240-241) Because this is not the first appearance.
- "DFS (mDFS: 30 vs. 18…." → "DFS (median DFS: 30 vs. 18…." (Page 8(4?), Line 241)
- "A higher pCR was also observed…." → "A higher pathological complete response (pCR) rate was also observed…." (Page 8(4?), Line 242)
- "in resectable esophageal or junctional adenocarcinoma [48]. " → "in resectable esophageal or junctional AC [48]." (Page 8(4?), Line 255) Because this is not the first appearance.
- "nor in mEFS (median event-free 278 survival: 28 vs. 25 months; HR = 0.99; p = 0.92)." → "nor in median event-free survival (mEFS): 28 vs. 25 months; HR = 0.99; p = 0.92)." (Page 9(5?), Line 278-279)
- "using the CPS or TAP score, has not yet…." → "using the CPS or tumor area positivity (TAP) score, has not yet…." (Page 16(2?), Line 504)
- "a greater disease-free survival (DFS) benefit…." → "a greater DFS benefit…." (Page 16(2?), Line5095) Because this is not the first appearance.
- "Fluorouracile + Leucovorin + Oxaliplatin (FOLFOX),…." → "Fluorouracil + Leucovorin + Oxaliplatin (FOLFOX), …." (Page 18(4?), Line 537, in Table 4 footnote)
- "Circulating tumor DNA (ctDNA) has emerged…." → "CtDNA has emerged…." (Page 19(3?), Line 555)
- "including diffusion-weighted MRI or…." → "including diffusion-weighted magnetic resonance imaging (MRI) or…." (Page 19(3?), Line 574) Or you should not abbreviate due to the single use in manuscript.
- "The integration of circu-601 lating tumor DNA (ctDNA) as a potential marker…." → "The integration of ctDNA as a potential marker…." (Page 19(3?), Line 601-602)
Response 1 : We sincerely thank the reviewer for this thorough and meticulous evaluation of our manuscript. We carefully reviewed all the suggested corrections related to terminology, spelling, and descriptive consistency, and we have implemented every proposed change throughout the text. These revisions have improved the overall clarity, precision, and consistency of the manuscript. We are grateful for the reviewer’s attentive reading and valuable attention to detail.
Round 2
Reviewer 3 Report
Comments and Suggestions for Authors
I have to confirm my previous review
Author Response
As discussed during our previous revisions, we have reaffirmed our rationale for including both adenocarcinoma and squamous cell carcinoma in the final manuscript. We have therefore decided to retain the discussion on SCC in the final revised and submitted manuscript.